# A Unified Sequence Interface for Vision Tasks

**Ting Chen**[†]    **Saurabh Saxena**[†]    **Lala Li**[†]
**Tsung-Yi Lin**[*]    **David J. Fleet**    **Geoffrey Hinton**
Google Research, Brain Team
{iamtingchen,srbs,lala}@google.com

## Abstract

While language tasks are naturally expressed in a single, unified, modeling framework, i.e., generating sequences of tokens, this has not been the case in computer vision. As a result, there is a proliferation of distinct architectures and loss functions for different vision tasks. In this work we show that a diverse set of "core" computer vision tasks can also be unified if formulated in terms of a shared pixel-to-sequence interface. We focus on four tasks, namely, object detection, instance segmentation, keypoint detection, and image captioning, all with diverse types of outputs, e.g., bounding boxes or dense masks. Despite that, by formulating the output of each task as a sequence of discrete tokens with a unified interface, we show that one can train a neural network with a single model architecture and loss function on all these tasks, with no task-specific customization. To solve a specific task, we use a short prompt as task description, and the sequence output adapts to the prompt so it can produce task-specific output. We show that such a model can achieve competitive performance compared to well-established task-specific models.

## 1   Introduction

Training a single neural network model capable of performing myriad tasks is a major step towards artificial general intelligence. In recent years, with the rise of big language models [34, 35, 2] using Transformers [41], many different language and related tasks are unified under a single modeling framework, where a language model is trained to predict the solution (in text tokens) given a prompt of a task description (also in text tokens). This is only possible because these tasks (both task description and solution) can be expressed in the same, rich language interface.

This can be naturally extended to some vision tasks such as image captioning or visual question answering where the solution is given in natural language, but the majority of "core" computer vision tasks have diverse outputs that are not readily expressed in terms of natural language. The object detection task produces a set of bounding boxes and their corresponding class labels, often associated with scores for ranking. The output for instance segmentation is a set of segmentation masks corresponding to image regions. The output of keypoint detection is a set of keypoints in an image. As such, existing methods [13, 37, 15, 28, 4, 15] have developed specialized architectures and sophisticated loss functions for each of these complex tasks.

An ambitious goal, in the pursuit of artificial general intelligence, is a simple interface that allows one to express seemingly disparate vision tasks in a unified framework. This would simplify the design of architectures and loss functions for new tasks. It would enable greater degrees of feature/representation sharing across many different tasks, thereby avoiding the need for a sophisticated

---

[†]Equal contributions. [*]Work done at Google.

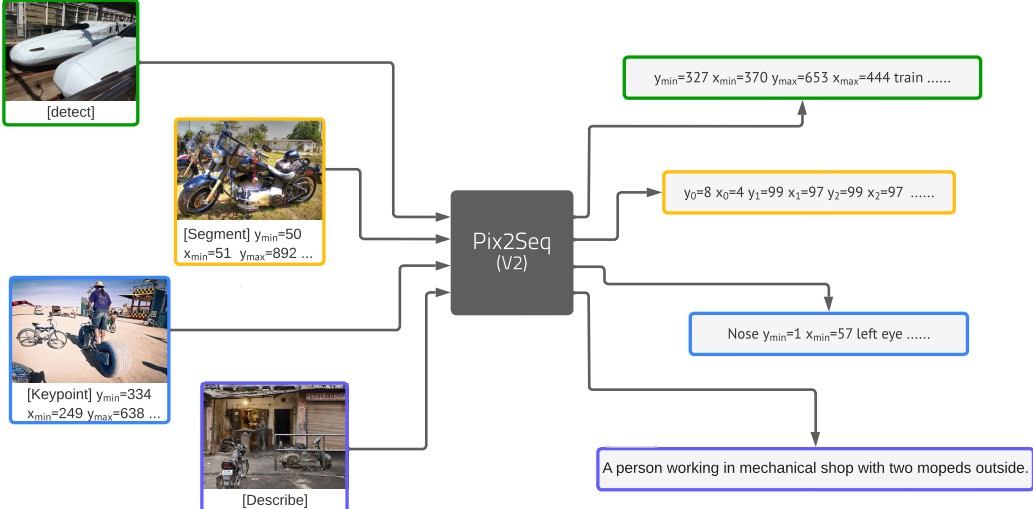

Figure 1: An illustration of the proposed framework. An image and a sequence of task prompt is given, the model produce a sequence of discrete tokens corresponding to the desired output.

output head for each task. It would also facilitate adapting of existing models to new tasks, and potentially unlock new capabilities with zero or few demonstrations.

To this end, we propose an approach to unify four seemingly different vision tasks in a single pixel-to-sequence interface. In effect, this is an extension of Pix2Seq [7] for object detection to a broader set of tasks. As a proof of concept, we focus on four core vision tasks, namely, object detection, instance segmentation, human keypoint detection, and image captioning. We first show how to unify these tasks into a single shared interface, and then train a neural network with a shared architecture and objective function. To solve a specific task, instead of using a specific head for that task, we use a prompt to specify the task, and the sequence output adapts to the prompt so it can produce task-specific output given the task description. This makes multi-task learning more efficient and scalable. We conduct experiments on the challenging COCO dataset, and show that it can simultaneously solve all four tasks well, without specialized architectures or loss functions.

## 2 Approach

In our approach, we cast computer vision tasks as one of translating pixel inputs (along with some descriptions of the task) into sequences of discrete tokens (see Figure 1). As a proof of concept, we focus on four core vision tasks: object detection, instance segmentation, keypoint detection, and image captioning; but we believe it is relatively straightforward to include many more tasks.

### 2.1 A unified interface with tokenization

The vision tasks we consider are diverse, and traditionally have been formulated quite differently. Object detection requires the model to produce bounding boxes for all objects without duplication. Instance segmentation requires the model to produce a dense pixel-wise mask for each identified object instance. Human keypoint detection requires the model to generate points corresponding to specific positions of landmarks on body parts for person instances (e.g., head, eyes). Image captioning requires the model to produce a sequence of words corresponding to a natural language description of the image. Given the significant differences in the form of the outputs, customized models with specialized architectures and loss functions are designed for each task.

To solve these tasks using a single model, we advocate the transformation/tokenization of task inputs and outputs into a unified interface. In this work, we propose a sequence interface for the purpose, where both task descriptions and outputs are expressed as sequences of discrete tokens:

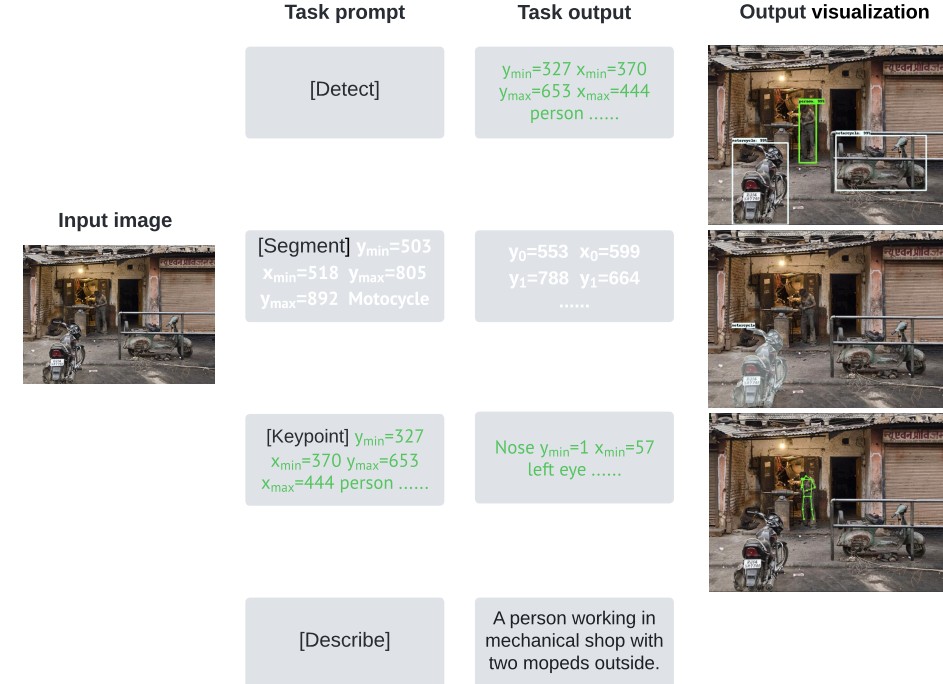

Figure 2: An illustration of sequence interface for four tasks. The model takes input image, task prompt tokens and produce task output tokens, which can be decoded/detokenized into required task output for visualization.

- For object detection, we follow [7] and convert bounding boxes and object descriptions into a sequence of discrete tokens by quantizing the continuous image coordinates. Specifically, an object is represented as a sequence of five discrete tokens, i.e. $[y_{\min}, x_{\min}, y_{\max}, x_{\max}, c]$, and multiple objects are randomly ordered each time a training image is sampled and serialized into a single sequence.

- For instance segmentation, instead of per-pixel masks, we predict the polygon [5] corresponding to the instance masks as a sequence of image coordinates conditioned on a given object instance. Again, we quantize the coordinates into discrete tokens. And to turn polygon into a sequence, we randomly select a starting point for the start token each time a training image is sampled. If there are multiple polygons for the same instance, we concatenate sequences of individual polygons with a separator token in between, so that every instance has a single corresponding sequence.

- For keypoint prediction, we predict a set of keypoints as a sequence of quantized image coordinates conditioned on a given person instance. Specifically, the sequence of keypoints can be encoded as $[y_{\text{keypoint 1}}, x_{\text{keypoint 1}}, y_{\text{keypoint 2}}, x_{\text{keypoint 2}}, \cdots]$. One may also use a keypoint label (e.g., nose, let eye, right eye) before each $(y, x)$-coordinates so their ordering does not need to be fixed but we opt for simplicity given that there are only a small fixed set of 14 person keypoints in the COCO dataset we consider. When certain keypoints are occluded, their coordinate tokens are replaced with a special occlusion token.

- For captioning, we directly predict text tokens given a caption is a sequence of discrete tokens.

It is worth noting that all four tasks share a single vocabulary. The specific prompts and output sequences are illustrated in Figure 2.

## 2.2 Unified architecture and objective function

We need a flexible and expressive architecture that can deal with image input and sequence output with complex semantics. Thus we follow [7] and use an encoder-decoder architecture, with an image encoder and sequence decoder. The image encoder perceives pixels and maps them into

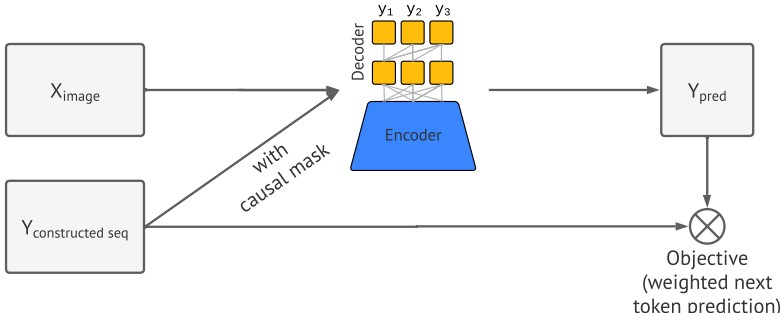

Figure 3: An illustration of our architecture and training objective. Note that $Y_{\text{constructed seq}}$ encapsulates both task prompt tokens and task output tokens. Token weights are set to zero if the target token is within the prompt so the model is only trained to predict desired output tokens.

hidden representations, which can be instantiated as a ConvNet [23, 22, 14], Transformer [41, 11], or their combination [4]. The Transformers-based sequence decoder, widely used in modern language modeling [41, 33, 35], generates one token at a time, conditioned on the preceding tokens and the encoded image representation. This removes the complexity and customization in architectures of modern neural networks for these vision tasks (such as per-task specific heads or necks [15, 19, 30]).

Unlike [7] where the decoder produces the output tokens directly for the single object detection task, here it also conditions on a task prompt so that the model can produce outputs adapted to the task of interest. During training, we concatenate both prompt and desired output into a single sequence, but leverage a token weighting scheme to ensure that the decoder is only trained to predict the desired output but not the prompt tokens. During inference, the prompt is given and fixed, so the decoder only needs to produce the rest of the sequence. Similar to [7], the training objective is to maximize the likelihood of tokens conditioned on an image and preceding tokens, i.e.,

$$\text{maximize} \sum_{j=1}^{L} \boldsymbol{w}_j \log P(\boldsymbol{y}_j | \boldsymbol{x}, \boldsymbol{y}_{1:j-1}) \, , \tag{1}$$

where $\boldsymbol{x}$ is the input image, $\boldsymbol{y}$ is a length-$L$ sequence associated with $\boldsymbol{x}$. As mentioned, the initial part of the sequence $\boldsymbol{y}$ is a prompt, for which we set the weight $\boldsymbol{w}_j$ to zero so it is not included in the loss.

## 2.3 Training

Each task has its own paired image-sequence training data. There are two ways one can combine tasks and perform the joint training.

**Data mixing.** We can create a dataset with mixed image-sequence pairs drawn from different tasks, balanced to account for different dataset sizes and task difficulties. This construction is extremely simple conceptually, but image augmentations can be difficult to incorporate as they may also require a change to their associated sequences in non-trivial ways.

**Batch mixing.** For each batch, we can sample images with annotations for a single task, perform image augmentations appropriate for this task, and convert the augmented data into image-sequence pairs. The model computes the loss and gradient for each task separately, and we can combine gradients from task-specific batches with an appropriate weighting.

---

**Algorithm 1** Training based on data mixing

1: Tokenize annotation into sequences of tokens,
2: Mixing images and sequences from all tasks,
3: Sample a batch, compute the loss, and update the model.

---

**Algorithm 2** Training based on batch mixing

1: Sample batches of data from all tasks,
2: Tokenize annotation into sequences of tokens,
3: Compute the loss for each task, aggregate their gradients, and update the model.

---

Algorithm 1 and 2 give summaries of both training strategies. We use batch mixing strategy in this work as it simplifies handling of image augmentations, but we are hopeful that data mixing

can be employed in the future to further simplify the pipeline and allow more tasks to be added straightforwardly.

Both data mixing and batch mixing require that we specify the portion or weighting for each task. This is an empirical matter, and we use a greedy strategy by adding one task at a time. Every time when we add a task, we adjust the weighting of the new task while keeping the relative weighting among the existing task fixed. We fix the sum of the weights across all tasks to be one.

## 2.4 Inference and de-tokenization

At inference time, we sample tokens from the model likelihood, given a prompt at the start of the sequence, i.e., $P(\boldsymbol{y}_j|\boldsymbol{x}, \boldsymbol{y}_{1:j-1})$. We currently use nucleus sampling [16] but other techniques such as beam search could also be used. Once the tokens are generated, they can be decoded for each task. In the same way that different tasks require specific tokenization schemes to generate token sequences, the decoding (de-tokenization) process is also specific to each task. A more detailed description of inference decoding for each task is given below.

- For bounding boxes, following [7], we split predicted sequences into tuples of 5 tokens to get coordinate tokens and a class token, and dequantize coordinate tokens to get the bounding boxes.
- For instance segmentation, we dequantize the coordinate tokens corresponding to each polygon, and then convert them into dense masks. The model is not trained with any geometry-specific regularizers per se, and as such the output polygonal masks can be somewhat noisy. To reduce the noise we find it helpful to sample multiple sequences and then average the masks, followed by a simple threshold to obtain a single binary mask.
- For keypoint detection, we directly dequantize the image coordinate tokens of the keypoints.
- For captioning, we directly map the predicted discrete tokens into text.

## 3 Experiments

### 3.1 Experimental settings and implementation details

We evaluate the proposed method on the widely used MS-COCO 2017 dataset [26], containing 118k training images and 5k validation images, spanning the four tasks we consider. An image in the dataset typically has annotations for object bounding boxes, segmentation masks for object instances, keypoint for person instances, and a few captions. Following [7], we use a Vision Transformer (ViT-B) encoder [11, 41], and a Transformer autoregressive decoder [41]. This model has a total of 132M parameters. To initialize the model, we use a pretrained checkpoint from [7] trained on the object detection task with the Objects365 dataset [39]; this is useful as COCO is relatively small and our model has less task-specific prior. For training on COCO, we use a batch size of 128 images, a learning rate of $1e^{-4}$, and we train the model for 100 epochs. We use a single vocabulary of 35K, with 32K text tokens, 1K coordinate quantization bins, and a few other class labels. We use a maximum sequence length of 512. Our backbone model is pretrained with 640×640 image size, and is fine-tuned in 640×640 or 1024×1024 resolutions.

**Object detection.** We follow [7] and use sequence augmentation during training, and use class token probability at inference time for scoring. We also use scale jittering as in [7] (scaling images randomly without changing aspect ratio, crop a fixed size region randomly, and then pad to the maximum size).

**Instance segmentation.** We set the maximum points of polygons to 128. We find that asking the model to generate multiple samples during the inference time and average the generated masks to be beneficial. More specifically, when multiple samples are independently drawn, we convert each of them into a semantic mask for the prompted object. We then average the masks by setting a (50%) threshold, and pixels with more than 50% times of being on will be selected for that instance. We find that 8 samples are sufficient to provide good performance (∼6 AP better than using a single sample), and beyond 12 samples we do not see performance boost. Additionally, during inference, we also evaluate on the cropped regions of the image containing the prompted object instance, by replacing the original input image with a new image only containing the cropped region. With smaller image size of 640×640, this yields 1.3 AP improvement, but with larger image size of 1024×1024, this does not seem to help much.

**Keypoint detection.** We train and evaluate on cropped regions of the image containing person instances (following the common practice in the community). During training, these regions are provided by ground-truth annotations, and during inference these regions are provided by the object detection model. We choose this region to be twice the size of the provided bounding box. We find that this works better than training with a larger crop size or cropping to the exact bounding box. Using our optimal crop we get ∼9 AP improvement over using an extremely large crop (∼20 times the box size which can be considered a close approximation to using the entire image). We also use a special token to represent invisible token coordinates in the quantized sequence. At training time we use a small loss weight of 0.1 for these tokens. While using a larger weight doesn't affect AP much (lower by 1 at weight 1.0) using a weight of 0.0 does much worse (12 AP lower). At inference time invisible tokens are replaced with the model's best guess of the keypoints' coordinates.

**Four-tasks joint training.** We use a mixed weighting of 0.1782, 0.7128, 0.099, 0.01 for object detection, instance segmentation, image captioning, and keypoint detection respectively. This set of weight is searched greedily by adding one task at a time (while keeping the weighting ratio of existing tasks unchanged). Ablations on task weighting are shown in the quantitative results below.

**Baselines.** We compare with a few well-known task-specific baselines. For object detection we compare with a strong 2-stage detector, Faster R-CNN [37], and a more recent Transformer-based detector, DETR [4]. Both Faster R-CNN and DETR use task-specific priors in their design, such as non-maximum suppression in Faster R-CNN and bipartite graph matching with generalized intersection-over-union in DETR. Due to their customized architectures and loss functions, extending them to a wider spectrum of tasks is non-trivial and may require a new model design. Mask R-CNN [15] advocates a design to extend Faster R-CNN to incorporate segmentation masks and keypoints. While Mask R-CNN is able to perform three out of our four tasks, it still requires the same set of task-based customizations as in Faster R-CNN. We also consider an improved version of Mask R-CNN with non-local architectures [43] which incorporates an attention mechanism, similar to Transformers. The above methods cannot do image captioning, so we train a Transformer-based caption model [40, 32] which is specialized for the task. This model is similar to the proposed method trained for caption single task but it is using self-supervised pretrained visual encoder [6] with a high dropout rate.

## 3.2 Quantitative results

Table 1: COCO results for object detection, instance segmentation and keypoint detection are expressed in terms of AP. For Image Captioning we report BLEU score. Single task results for instance segmentation and keypoint detection are based on detected bounding boxes from single task detection model. - indicates the model is not able to solve the task without modifications.

|  | Object det. | Instance seg. | Keypoint det. | Captioning |
|---|---|---|---|---|
| Faster R-CNN [37] | 42.0 | - | - | - |
| Faster R-CNN+ [37] | 44.0 | - | - | - |
| DETR [4] | 44.9 | - | - | - |
| Mask R-CNN [15] | 39.8 | 37.1 | 63.1 | - |
| Mask R-CNN (non-local) [43] | 45.0 | **40.3** | 66.5 | - |
| Transformer-based captioner [41, 32] | - | - | - | 34.3 |
| Pix2Seq v2 single task (640×640) | 43.8 | 37.3 | **68.0** | 33.9 |
| Pix2Seq v2 single task (1024×1024) | 45.6 | 38.7 | 67.4 | 34.0 |
| Pix2Seq v2 multi-tasks (640×640) | 44.2 | 36.9 | 65.0 | 34.3 |
| Pix2Seq v2 multi-tasks (1024×1024) | **46.5** | 38.2 | 64.8 | **34.9** |

Our main results are summarized in Table 1, where we report baselines and two variants of our model: (1) single task models where the model is trained on a single task (still with the same architecture and objective function), so each task has its own network weights; and (2) a multi-task model, where a single set of network weights is used for all four tasks. We can see that despite without task-specific priors in architecture and loss function, our model can still achieve competitive results for each individual task compared to strong specialized baselines (even with a smaller image size). When we train a single model on all tasks, our model is able to address these tasks relatively well, despite the model size being kept the same. We also observe that, with larger image sizes, the performances are

generally improved. One exception is keypoint detection, which already uses a cropped region of interest for detecting key points, thus scaling up the image size is not necessarily helpful and can lead to overfitting in case of limited labeled data.

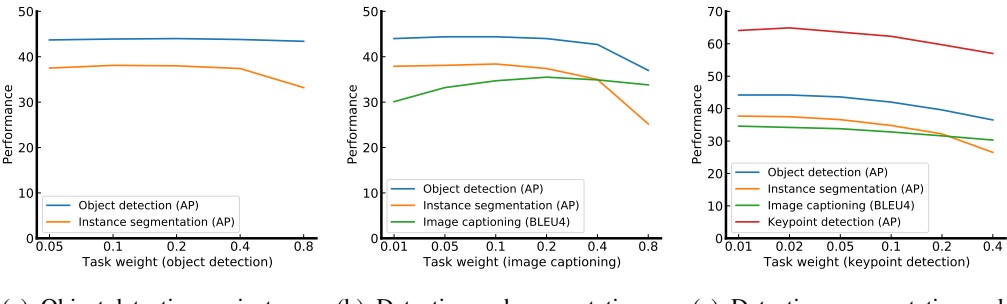

(a) Object detection *vs.* instance segmentation.

(b) Detection and segmentation *vs.* image captioning.

(c) Detection, segmentation and captioning *vs.* keypoint detection.

Figure 4: Performance with different task weighting when a new task is added into an existing task mixes.

Figure 4 shows how we select appropriate loss weighting for each task using a greedy strategy. More specifically, we first search the weight ratio between object detection and instance segmentation and results are shown in Figure 4a. We observe that for a relatively wide range of weighting ratios, the performance of both tasks are near their peak, so we simply choose the 2:8 weighting ratio for these two tasks. After that, we add image captioning task, and the performances under different weighting of the captioning task can be found in Figure 4b, where we find that the 9:1 weighting ratio for existing tasks and image captioning tasks to be appropriate. Finally, adding the keypoint detection task, in Figure 4c we find its weight can be set relatively small and we choose to use 0.01.

### 3.3 Qualitative results

To demonstrate the capability and performance of our model in a more visual and intuitive way, we show the outputs from our multi-task model on selected images from the COCO validation set, for each of the four tasks, i.e., object detection, instance segmentation, keypoint detection, and image captioning. Figure 5 shows results for the object detection task. The model successfully detects objects of different sizes in cluttered scenes with significant occlusion. Empirical results on instance segmentation and keypoint detection are shown in Figures 6 and 7. For both tasks, the multi-task model produces well localized and accurate predictions. We also demonstrate some captions generated by the model in Table 2. With these results, we note that our model has not been pre-trained using large-scale image-text datasets, which is expected to significantly improve the captioning performance of the model.

## 4 Related work

**Decoding visual concepts:** Image understanding involves extracting visual concepts from images. The formats of these concepts vary according to the given task. Image captioning uses a sequence of words to describe an image [9]. Object detection, on the other hand, represents objects with labels and bounding boxes. Depending on the granularity of localization, visual concepts can be expressed as boxes, pixel segmentation, or keypoints [26]. Decoding localized visual concepts often requires tailored methods. For example, image segmentation uses per-pixel classification. Object detection uses sliding window with non-maximum suppression to detect boxes. Person keypoint detection uses part models to assemble detected parts into whole body [3].

Recently, DETR [4] is proposed as an end-to-end object detection approach based on a Transformer decoding scheme (removing complexity on bounding box proposal and non-maximum suppression). MaskFormer [10] further shows that object detection and segmentation can share the same decoding scheme. Pix2seq [7] demonstrates that boxes and labels can be treated as a sequence of discrete tokens, thereby sharing the same training and decoding interface as language models [33, 35]. In our

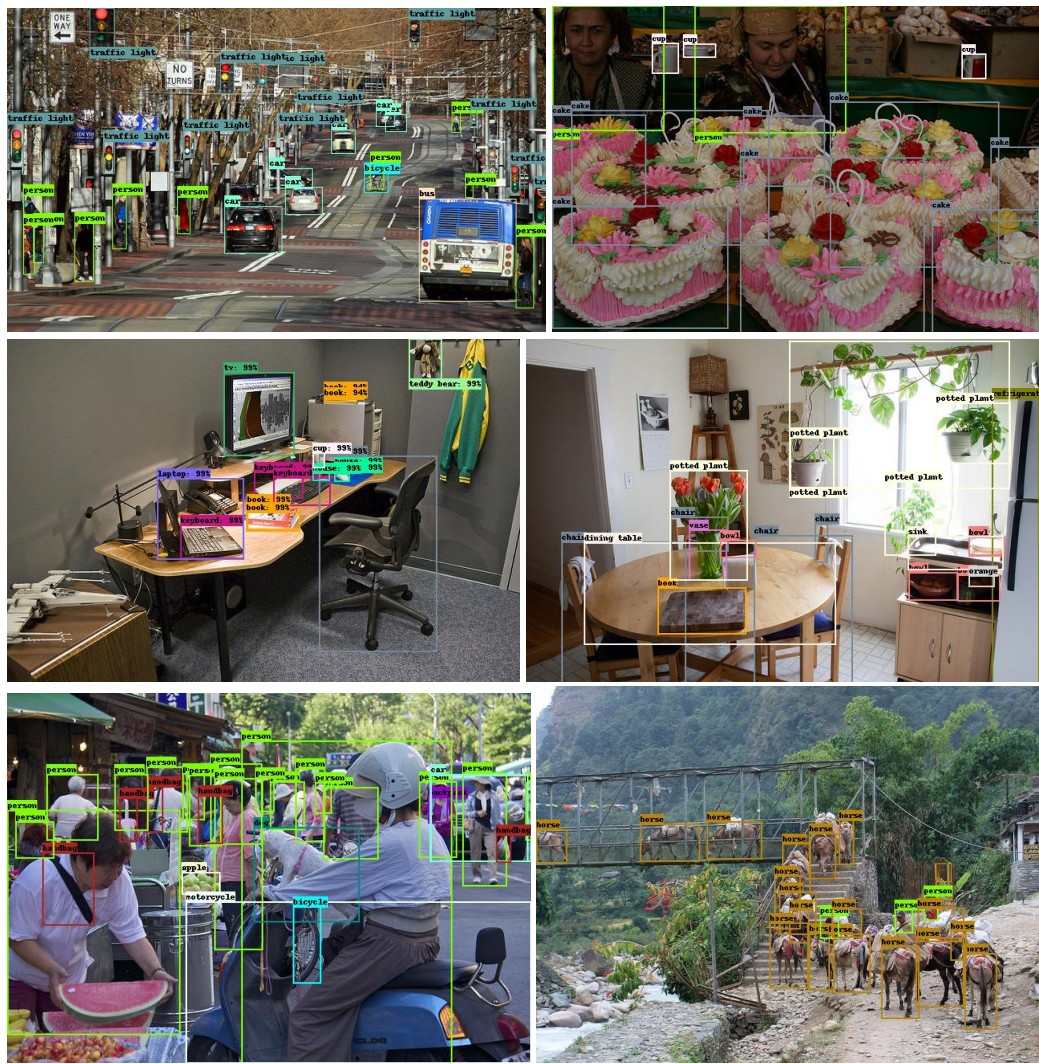

Figure 5: Visualization of the object detection results, with predicted bounding boxes on the input images.

work, we push the envelope further in the unification of language and different visual localization tasks to share the same interface, architecture and training objective.

**Generalist vision models:**   Learning a generalist model capable of performing multiple tasks is widely assumed to be a path toward general intelligence. In visual recognition, multi-task learning has shown great success by sharing a backbone model, followed by multiple independent heads [15, 19, 30]. Models with a shared backbone can learn general features which are transferable across tasks when scaling up with training tasks, model capacity and data [20, 24, 12]. Nevertheless, often the task specific backbone models are designed carefully, particularly for tasks that require accurate localization [38, 27].

With the invention of Transformers [41], recently being adopted for image classification [11], the research community has seized on the opportunity to unify the backbone design for vision tasks [29, 8, 25]. Perceivers [18, 17] and OFA [42] demonstrate an architecture for multi-task and multi-modal across vision and language. Notably, OFA designs a unified sequence-to-sequence decoding architecture for both language and object detection tasks. Flamingo [1] and related methods also focus on an universal API that produces a natural language output for a variety of tasks given image input. This line of work shares a common motivation to our work in this paper, however they focus on higher level tasks for which natural language is inherently the desired output. In this paper we

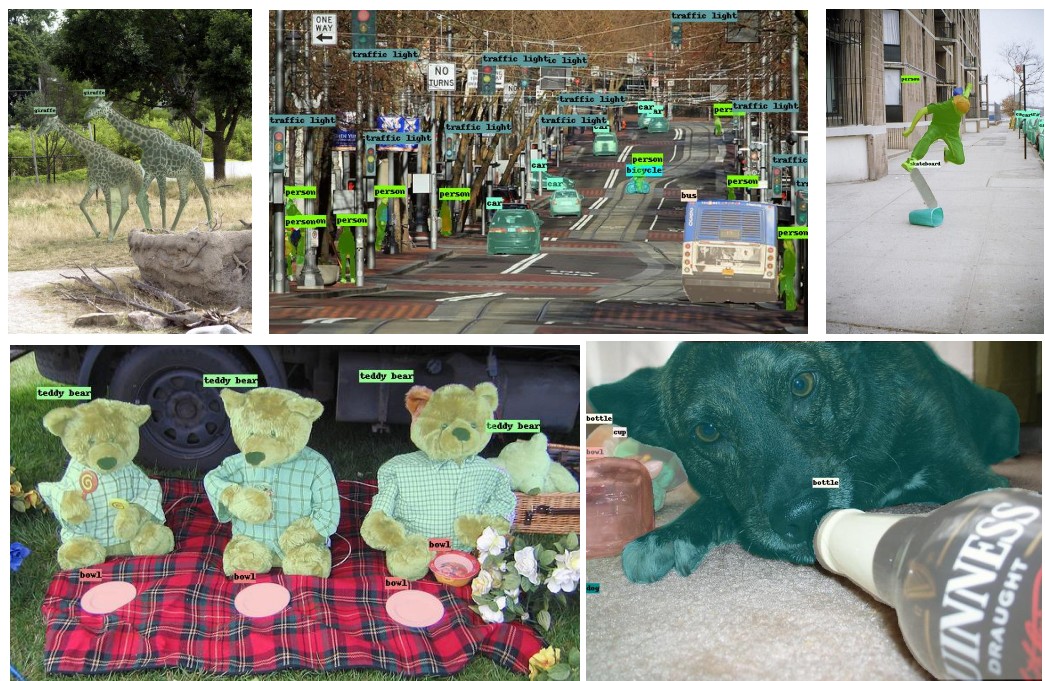

Figure 6: Visualization of the instance segmentation results, with predicted semantic masks overlaid on the input images.

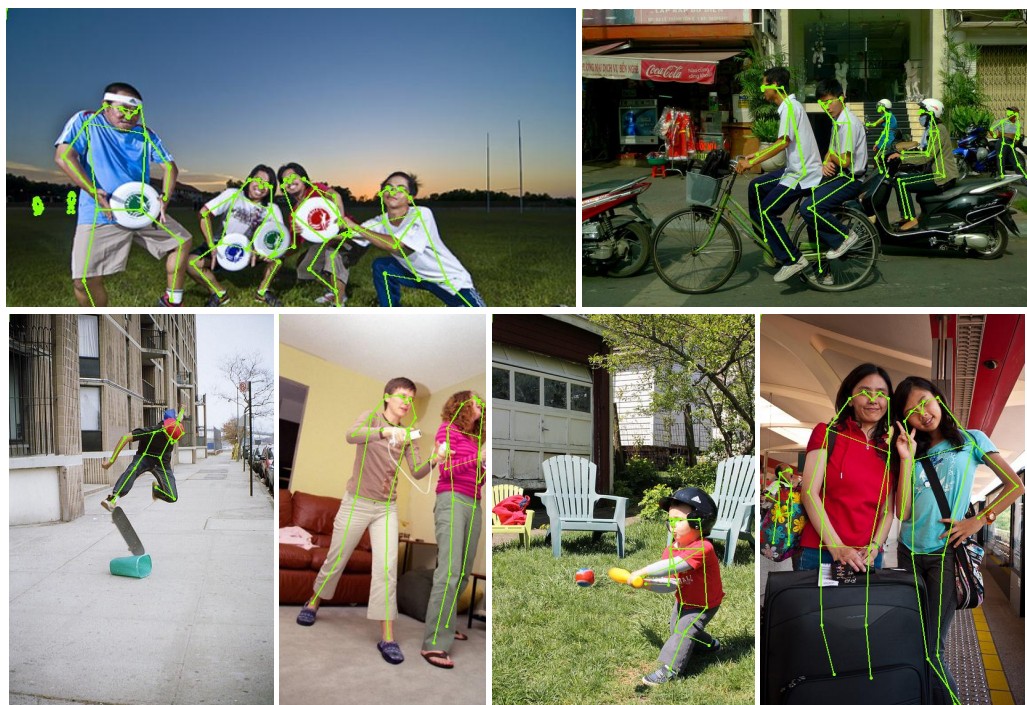

Figure 7: Visualization of the Human keypoint detection results, with predicted stick figures on the input images.

Table 2: Image captioning results.

| | |
|---|---|
| 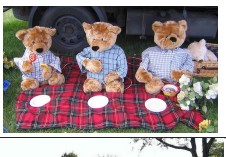 | A group of teddy bears sitting next to each other. |
| | Three teddy bears sitting on a blanket. |
| | A group of teddy bears sitting on a blanket with bowls of food. |
| 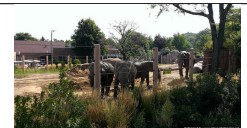 | A herd of elephants standing inside of a fenced in area. |
| | A group of elephants standing in a fenced area. |
| | A herd of elephants standing behind a fence. |
| 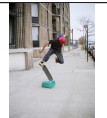 | A man riding a skateboard over a block of cement. |
| | A man doing a trick on a skateboard in the street. |
| | A man flying through the air while riding a skateboard. |
| 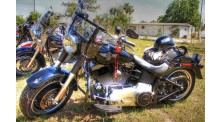 | A row of motorcycles parked on a grass covered field. |
| | A motorcycle with a helmet on the side of it. |
| | A motorcycle parked in a grassy area with other motorcycles. |

demonstrate that one can express a variety of "core" computer vision tasks in a universal interface, and the learned model exhibits strong grounding capability of the tokens they produce to actual visual concepts. Concurrently to our work, Gato [36] unifies a series of vision and control tasks into a single sequential prediction problem, and UViM [21] and Unified-IO [31] propose using learned discrete codes for unifying a set of vision tasks.

## 5  Conclusion

In this work, we explore a unified sequence interface for tackling a diverse set of "core" vision tasks, where both the task description (prompt) and task output are expressed as discrete sequences of tokens. This is a significant departure from conventional norms of multi-task vision models in that both architecture and loss functions are shared among the tasks. We show that such a model can achieve competitive performance compared to well-established task-specific models.

Our work is not without limitations. Due to the significant departure from conventional approaches, we believe both architectures and other training techniques can be further improved to challenge the state-of-the-art of specialized systems. We also believe our model can significantly benefit from scaling up, both in pretraining on larger datasets (e.g., image-text pairs) and/or using larger model sizes. Another limitation is the inference speed can be potentially slower (for longer sequences particularly) compared to the specialized systems as our approach is based on autoregressive modeling. There are a few ways to improve the efficiency, including using non-autoregressive sequence modeling (which we leave as future work). In this work, we exploit parallel querying for speeding up our model inference. For example, predicting multi-person poses can be done independently by prompting the model with independent bounding boxes (detected by the model itself or pre-given), so the only sequential prediction is limited to a single person with a few keypoints. The same strategy can be applied to instance segmentation as well.

While the optimal implementation of a unified interface still requires more research and the sequence interface explored in this work is only one potential implementation, we believe the interface of how different tasks are formulated would play an increasing important role in general-purpose intelligent systems going forward.

## Acknowledgements

We specially thank Wei Li for their helpful feedback on the initial draft. We also thank Xiaohua Zhai, Alexander Kolesnikov, Lucas Beyer, Neil Houlsby, Simon Kornblith and Mohammad Norouzi for some early discussions.

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
