# OpenReview forum: "A Unified Sequence Interface for Vision Tasks"
_NeurIPS.cc/2022/Conference — NeurIPS 2022 Accept_

### Official Review · Reviewer_vjx3 · 2022-07-10

**Rating:** 5
**Confidence:** 5
**Soundness:** 3 good
**Presentation:** 2 fair
**Contribution:** 2 fair

**Summary:**

This paper addresses the unification of 4 specific vision and vision-language tasks (namely object detection, instance segmentation, human pose estimation, and image captioning) with a transformer framework and a sequence modeling objective. The experiments show that their method achieves competitive results compared to well-established task-specific models.

**Questions:**

see weakness

**Strengths And Weaknesses:**

Strengths:

1. This paper follows the promising trend of task-unification under a transformer framework with sequence modeling, and the authors extends Pix2Seq model to learn 4 specific tasks in COCO datasets.
2. The proposed method simplifies the training objective and model architecture design.
3. With the nature of transformer architecture and seq. modeling, it allows text-prompting to activate desired functionality (although the text prompt is pre-defined). This helps create a friendly user interface to access the model.
4. Although the proposed model does not establish new state-of-the-art performance across the 4 specific tasks, the results are still very competitive. In my opinion, the results are promising as it could initiate a new research line unifying different computer vision tasks. The proposed model shows great potential for the ambitious goal of the all-in-one design; however, I also have several concerns about this paper, and they are listed below.

Weaknesses:

In NLP and vision-language fields, a single sequence modeling objective shows great advance as most of their tasks can be easily formulated as a sequence prediction task. However, for computer vision tasks, it remains unclear whether most of the vision problems can be solved with such a sequence modeling. While the authors demonstrated promising results on 3 selected vision tasks and 1 vision-language task, many other vision problems developed in the community are not clearly addressed. While it is really an open research question, the title of this paper "A Unified Sequence Interface for Vision Tasks" is indeed overclaimed. I would suggest rewrite as "A Unified Sequence Interface for COCO Dataset" to better reflect what you have proposed and what you have done in the paper.

To be more specific, the authors claimed that they consider "core vision tasks", but task definition seems bias as it is unclear what should be core, and which should not. In my view, there are only 3 pure vision tasks (detection, segmentation, human pose estimation), and additionally the authors consider a vision-and-language (V+L) task (i.e, image captioning). This introduces many concerns. It is unclear why the authors didn't include other popular V+L tasks such as image-text retrieval and visual question answering as they are commonly evaluated in the foundation model literature. Similarly, there are many popular pure vision tasks ignored in this paper, such as image/video recognition, NeRF, etc. Overall, this gives the impression that the task unification is achieved just because of a careful selection of downstream tasks in the experiments.

The authors compare their method with MaskRCNN, and show that their proposed method can additionally work for image captioning. But no comparison with V+L foundation models is provided. Since image captioning has been well-studied in the unification literature in V+L field, the main new thing here is probably about unifying 3 selected pure vision tasks. However, pioneer works (such as MaskRCNN, HRNet, and others) already unify the three tasks very well. It is unclear what is the advantage of the proposed method (i.e., using the sequence modeling for the three tasks).

From modeling perspective, it remains challenging for transformers to learn fine-grained spatial relationship, and such spatial info is critical to the selected 3 tasks. The motivation of using sequence modeling for the three tasks is somewhat weak.

From the perspective of efficiency, the proposed unification removes the need of task-specific modules design for COCO dataset. However, due to the nature of autoregressive prediction, the proposed model will be slow. For example, the bottom-up approaches in human pose estimation literature can perform both multi-person pose estimation and instance segmentation at the same time with just a single forward-pass, but in this paper, the inference cost could grow linearly to the number of people. The inference interface is kind of restricted: (1) only a single task is supported at a time; (2) it predicts token-by-token slowly.

From the perspective of training, the proposed method doesn't show great benefit. In NLP and V+L research, people use a single loss function, so that it is easier and scalable when performing large-scale pre-training. However, in this paper, the tasks considered require fine-grained annotations, and thus they do not have large-scale pre-training data (except for image captioning). For the considered downstream dataset, it is unclear if single-objective is more effective than multi-objective training.

Overall, the model design is neat and elegant. But it is not well-grounded to the considered downstream tasks. This paper is more like a system paper integrating a few known components together to achieve an expected "okay" performance. Considering the technical novelty, the contribution of this paper is relatively weak.

The proposed method may bring significant advantage with large-scale pre-training. But given the current paper content, it is difficult to reach a conclusion at this point.

---

> ### Author Response · Authors · 2022-08-02
> **Response to Reviewer vjx3**
>
> We thank the reviewer for their thoughtful feedback, please find our responses below.
>
> **The selection of vision tasks**
>
> Existing vision-language (image-text) modeling research has focused mostly on high level visual tasks, such as image/text retrieval, and question answering. Their modeling is mostly based on contrastive learning (sometimes they also incorporate image captioning), but they cannot solve mid-to-low level tasks such as detection, segmentation or keypoint prediction. In this work, as a proof of concept, we select a diverse set of representative tasks such that 1) the outputs can be expressed compactly and communicated through a low-bandwidth channel, and 2) they are widely used and studied across the vision community, and 3) they achieve a good balance across low, medium and high level visual understanding. We apologize if the title or our writing in any way misled you to believe that we solve all vision tasks, but we have made it clear throughout the paper that we have only tackled four vision tasks that we think are representative, or at least play an important role for proof-of-concept. Furthermore, although we only include four tasks, they have strong connections with other mentioned tasks, such as image image/video recognition (a special case of object detection without bounding box or bounding box being the whole image/video). We do not consider 3D tasks or image/video generation tasks (like NeRF), nor do we suggest in the paper that we do.
>
> **Efficiency**
>
> We acknowledge that autoregressive prediction is dependent on the sequence length and can be slower for predicting long sequences. However, there are a few ways to improve the efficiency, including using non-autoregressive sequence modeling (which we leave as future work). In this work, we exploit parallel querying for speeding up our model inference. For example, predicting multi-person poses can be done independently by prompting the model with independent bounding boxes (detected by the model itself or pre-given), so the only sequential prediction is limited to a single person with a few keypoints. The same strategy can be applied to instance segmentation as well. Therefore, we can easily support multiple people/task inference by parallel prompting. We will revise the paper to clarify this point.
>
> **Benefits of unified interface and single model training**
>
> The reviewer also asks about the benefits of a unified interface/model. Specialized systems can work very well in a pre-defined domain, but they can be difficult to generalize to new tasks (without specific modification of model and re-training). A unified approach may look worse in performance at the beginning (compared to specialized systems), but it takes time to improve them, and we believe it is a necessary step towards a general intelligence system that can interact with and help people in myriad  ways. An important milestone toward a truly usable generic system is to have the architecture and loss function be  general, which was our goal in this work.  If a (new) task cannot be expressed in the system’s interface and trained with one’s architecture/loss, the system has no chance of generalizing it. In other words, without a unified interface, there is zero chance a model is able to generalize to diverse sets of new tasks - they simply cannot represent them! People may not agree if the specific approach proposed in this work is the right way to go, but we believe it is worthy of exploration. And future work can present new approaches for the interfaces or new interfaces, but the early explorations should not be regarded as not useful.

---

> > ### Comment · Reviewer_vjx3 · 2022-08-08
> > **more comments**
> >
> > Thanks for the detailed feedback.
> >
> > Since it is not trivial to apply seq. modeling to instance segmentation and keypoint detection, it is very possible that it cannot be directly applied to all the other vision tasks. It remains an open question how to apply it to other vision tasks.  As I suggested in my original post: (1) I think the unification is achieved because of a careful selection of downstream tasks; (2) Current title of this paper is overclaimed. I see the authors clarify it throughout the paper, and mentioned it is just a proof-of-concept. However, if there is no overclaim, no clarification is needed at all, right? If the proposed method is not ready to address all vision tasks, it seems not suitable to called "A Unified Sequence Interface for Vision Tasks". The title seems too strong considering the paper content.
> >
> > Regarding efficiency, it is unclear what's the difference between parallel computing and the proposed possible solution. If parallel computing/prompting is allowed, it just improves the throughputs of the existing models. It is technically valid, but it doesn't address the fundamental issue in the proposed model: (1) only a single task is supported at a time; (2) it predicts token-by-token slowly. It seems like existing multi-task vision system like MaskRCNN are still more efficient if parallel computing is allowed.
> >
> > I have mixed feeling regarding the proposed method. I believe this is a right direction to pursue, but overall, the novelty is somehow weak. The paper is more like integrating a few known components together. If it is non-trivial to put the selected 4 tasks together in a seq. modeling framework, it also implies it is non-trivial to add other new tasks. I think the paper somewhat lacks ingredients that can automatically adapt new tasks.

---

> > > ### Author Response · Authors · 2022-08-09
> > > **response**
> > >
> > > Thanks for the comments, please find our response below.
> > >
> > > **Since it is not trivial to apply seq. modeling to instance segmentation and keypoint detection, it is very possible that it cannot be directly applied to all the other vision tasks.**
> > >
> > > Tasks can be categorized and each category has some representative tasks. The tasks we selected are a representative set of vision tasks (this is why they are selected in major dataset such as COCO). Showing how to make sequence modeling work for these representative tasks would make other vision tasks (that they represent, or can be reduce to the representative tasks) much easier.
> > >
> > > **If it is non-trivial to put the selected 4 tasks together in a seq. modeling framework, it also implies it is non-trivial to add other new tasks. I think the paper somewhat lacks ingredients that can automatically adapt new tasks.**
> > >
> > > Again, these tasks are non-trivial because they are representative, and addressing these four tasks indicate the model's ability to go beyond. It is always difficult to have a new and unorthodox approach compared to minor tweaks of established approach, as existing work has built in years of tuning and refinement, but it doesn't mean the new approach wouldn't evolve and become better/easier over time. Please note that we did not claim we can automatically adapt new tasks in this work, this is a promising future, and our work (showing it works for representative tasks) is a step stone towards that direction.
> > >
> > > **The title seems too strong considering the paper content.**
> > >
> > > We respectfully disagree with this assessment as in no way we indicate our approach can solve *all* vision tasks in the paper. It is just *an* approach, and for *multiple tasks*. While we appreciate the reviewer’s suggestion of “A Unified Sequence Interface for COCO Dataset”, we do not think it’s an appropriate title for our paper since we are considering general vision tasks, not a specific dataset. We evaluate on the COCO dataset because all our baselines do, as COCO is the dominant dataset for these representative vision tasks. We remain open to other suggestions to avoid confusion.
> > >
> > > **It seems like existing multi-task vision system like MaskRCNN are still more efficient if parallel computing is allowed.**
> > >
> > > Our approach is dramatically different from MaskRCNN (which built on years of research of RCNN, faster RCNN and so on), so the efficiency difference can be tolerable for now. We acknowledged the efficiency limitation of our current system compared to these existing and more mature systems in the paper. Again, we urge the reviewer to keep an open mind for new approaches as they will be improved over time. One can’t achieve general visual intelligence with MaskRCNN, one may have a (small) shot with the proposed approach but it is a long way to go.

---

### Official Review · Reviewer_iNjL · 2022-07-11

**Rating:** 5
**Confidence:** 3
**Soundness:** 3 good
**Presentation:** 3 good
**Contribution:** 3 good

**Summary:**

This paper presents a unified sequence-to-sequence generation interface for 4 core vision tasks: object detection, instance segmentation, keypoint detection and image captioning. The model is designed with a single encoder-decoder architecture, with no task-specific heads. To differentiate among different tasks, descriptive task prompts are added as the starting token to generate the output sequence. Experimental results demonstrate a proof-of-concept that such a unified interface can achieve competitive performance on 4 tasks evaluated, compared to some well-established task-specific models.

**Questions:**

- Results on training from imagenet-pretrained weights? This can be important to understand the current advantage over the compared baselines. The baselines seem to be all trained from imagenet-pretrained weights?
- More details on the segmentation inference strategy. How the multiple samples are generated during inference? What kind of sampling strategy are used? How was the threshold decided?
- Any insights on the choices of  loss weights, as greedy search may not be feasible when scaling up the number of tasks.

**Limitations:**

Yes. The authors have discussed the limitations and societal impacts in Section 5.

**Strengths And Weaknesses:**

- Strength

In general, the paper is clearly written, easy to follow. Experimental results indeed show a proof-of-concept that this is possible to tackle vision tasks with a unified interface.

- Weakness

One concern I have is that the only technical contribution of this paper seem to be extending pixel2seq to the 3 other vision tasks.

- Minor comments

The important details on training and inference may be better to put into the main paper.

---

> ### Author Response · Authors · 2022-08-02
> **Response to Reviewer iNjL**
>
> We thank the reviewer for the feedback, please find our responses below.
>
> **novelty**
>
> Our work is built upon pix2seq, so the framework is the same and fundamental ideas (e.g., architecture and loss) are similar.  But this does not mean this work is incremental or inconsequential. Notably, the original pix2seq work only demonstrates the sequence modeling works for object detection, a single vision task. In this work, we propose concrete extensions and demonstrate that the framework can work with a very diverse a set of selected, representative vision tasks, covering low, mid, and high-level visual understanding.
>
> In particular, extending original pix2seq to image captioning is straight-forward, but it is not trivial to apply it to instance segmentation (which requires dense semantic masks), or to keypoint detection. More importantly, we solve challenges in joint training of multiple tasks and demonstrate the same sequence modeling framework can lead to competitive results in multiple “representative/core” vision tasks. As an example, one may say GPT-n is just incremental on top of GPT-1 for the reasons that they all use the same autoregressive transformer model to train language models of different sizes, but clearly they are not incremental. We believe this work is an important step toward a unified model for computer vision.
>
>
> **Pretraining and comparison with baselines**
>
> Our model is pretrained using Object365, and the baselines are pretrained on ImageNet. The main reason we pretrain on Object365 is that autoregressive Transformer decoder cannot be pretrained using ImageNet (since ImageNet does not contain object detection annotations). This is not an issue for the  baselines in our paper because their decoders / heads are localized and specialized, thus do not necessarily require substantial decoder pretraining. Both Objects365 and Imagenet are similar in size with over a million images, but are annotated differently. Objects365 has only 365 objects annotated, while Imagenet has 1000 classes of objects annotated. It is not entirely clear how they compare in terms of serving as a general pretraining dataset. We leave this study to future work.
>
> We also wanted to emphasize that our goal is not to beat the state-of-the-art in this single piece of work. Existing specialized systems have many years of research (tricks and optimization) built in, so it is not always possible to beat SOTA with an entirely new approach in a single try. The main message from our comparison is that: even compared to these very strong baselines (SOTA as of a few years ago), our new approach, without specialization used in those techniques, can still achieve a similar level of performance, serving as a proof of concept and showing a promising direction.
>
> **Segmentation inference**
>
> We first generate multiple *independent* polygons using nucleus sampling with top_p=0.8 (due to the nucleus sampling, the independent samples are relatively diverse). We then convert  each polygon into an instance mask (i.e. a binary mask, where pixels with 1 means it belongs to the instance). We can average multiple binary masks to get the mean-mask (pixel values between 0 and 1).  Thresholding the mean-mask then converts the mean-mask into the final binary instance mask. Both the number of samples and threshold can be determined by a validation set, but we find the performance is  robust to a wide range of parameters (e.g., more samples are better, but the performance saturates around 8).
>
> **others**
>
> *Any insights on the choices of loss weights, as greedy search may not be feasible when scaling up the number of tasks.*   One major consequence of bad weighting seems to be setting high weights for easy tasks so that they overfit significantly, so this may be used to come up with some heuristics to mix tasks (similar to how researchers mix different corpura/datasets in NLP). We also expect that, with large-scale pretraining (e.g., using image-text pairs), the task weighting would be less consequential.
>
> *The important details on training and inference may be better to put into the main paper.*   Well noted.  We will move  more details into the main body of the paper.

---

> > ### Comment · Reviewer_iNjL · 2022-08-09
> > **Response**
> >
> > Thanks the authors for detailed response. After reading the rebuttal and other reviewer's comment, I share the same concerns with Reviewer vjx3 about the novelty and how it can be extended to other tasks. In addition, as the authors pointed out in rebuttal, it is not entirely clear how to compare object365 and imagenet as the pre-training dataset and the SOTA compared in this paper is from a few years ago. Hence, I am still not convinced about the comparison to these models. Especially, when we are not sure whether the pre-training corpus contributed to the comparable performance.

---

> > > ### Author Response · Authors · 2022-08-09
> > > **response to response**
> > >
> > > Thanks for the follow-up, we appreciate it. We fully agree that a paper needs novelty to be worthy of publication, but please note that novelty does not equal a new technique. We have tens of thousands of published paper every year, but we do not really need tens of thousands of new techniques invented every year. Our novelty is to show how sequence modeling can tackle a set of representative vision tasks, and this has never been shown before. This result itself is novel (along side some specific modeling designs which we choose not to emphasize). We do not try to claim our approach can do better than SOTA specialized vision systems, as we just try to say "Look, here is a new generic (task-agnostic) approach based on a unified sequence model that can do pretty well for these set of representative tasks". This claim/message is somewhat orthogonal to the use of ImageNet pretraining or Object365 pretraining, as we only use the result numbers to show that the proposed model is capable of performing at a reasonably good level, not to say we are better than baselines (in performance). That said, we do believe the new approach can be improved over time, and our work is an important step stone to that direction. One can’t achieve general visual intelligence with existing specialized systems (despite they have very cool applications and positive impacts on society), one may have a (small) shot with the proposed approach but it is a long way to go.

---

> > > > ### Comment · Reviewer_iNjL · 2022-08-09
> > > > **Response**
> > > >
> > > > Thanks for the last minute reply. I agree that the authors have made some fair points in the last response. I am willing to update my score to boarderline accept.

---

### Official Review · Reviewer_KcLQ · 2022-07-11

**Rating:** 5
**Confidence:** 4
**Soundness:** 2 fair
**Presentation:** 2 fair
**Contribution:** 2 fair

**Summary:**

The work proposes an unified framework/interface for multiple vision and vision+language tasks, and achieves comparable performance on multiple tasks compared to other strong baselines. The work also shows benefits of multitask learning on some of the tasks. There are some concerns about novelty and whether the work is just incremental compared to previous work of pix2seq[6]. In addition training(pretraining) setup is slightly confusing and performance on few tasks are not so encouraging when compared to other state-of-the-art works.

**Questions:**

- How are the batches for different tasks ordered? Does that have any effect on the performance? Does curriculum learning help here?  If we order the batches for different tasks in a particular sequence, does their learning behavior and performance change? How do the training/validation plots look like? Does any task overfit when other tasks are still improving?
- How does multitask performance vary if we pretrain the model on other tasks than object detection? Is there a relation between high performance on obj detection in the unified setup to the fact that the model was initialized from pretrained weight for that task; however the other vision only tasks do not show good performance.
- How does the current setup compare to other works in terms of training/inference efficiency?
- What if the multitask trained model is again finetuned on individual tasks? Does multitask training as a pretraining step help?


**Limitations:**

Yes the authors have touched upon the limitations of this work. However I would encourage the authors to add more details about any potential negative societal impact this type of work can have.

**Strengths And Weaknesses:**

### Strengths

- The proposed unified interface is simple and beneficial for a multitask learning.
- Unifying the interface for these tasks enables same input and output for all tasks which makes it easily scalable to multiple tasks.
- Multitask learning shows improvements on some of the tasks.

### Weaknesses

- Authors say the models are initialized from pretrained checkpoint trained on the object detection task with the Objects365 dataset. Are the other baselines that are compared in Table 1 also initialized from pretrained object detection checkpoints on Objects365? If not then this is not a fair comparison.
- The work seems to be incremental and just adding more tasks on top of previous work of pix2seq[6]. If the main message of this work is multitask learning, then it needs to be highlighted properly and the benefits of doing multitask with a unified framework needs to be analyzed in detail. What is the advantage of this setup compared to other frameworks that have task specific input and output parameters/heads and a common  shared base for all tasks? More specifically the work should ablate why a unified interface important compared to architectures having task specific input/output heads.
- Multitask learning for Instance Segmentation and Keypoint detection are not helping compared to specialized single task models. The work lacks details and analyses for the potential causes.
- Authors mention that one benefit of this work is that the loss function can be shared. Why is that? The work misses ablation study on the benefit of a common loss function.

---

> ### Author Response · Authors · 2022-08-02
> **Response to Reviewer KcLQ**
>
> We thank the reviewer for the feedback.  Below, please find our responses to the questions and issues raised in the review:
>
> **Pretraining and comparison with baselines**
>
> We acknowledge that the our model is pretrained using Object365 while the baselines are pretrained on ImageNet. The main reason we pretrain on Object365 is that autoregressive Transformer decoder cannot be pretrained using ImageNet (since ImageNet does not contain object detection annotation sequences). This is not an issue for the baselines in our paper because their decoders / heads are localized and specialized, thus do not necessarily require substantial decoder pretraining. Both Objects365 and Imagenet are similar in size with over a million images, but they are annotated differently. Objects365 has only 365 objects annotated, while Imagenet has 1000 classes of objects annotated. It is not entirely clear how they compare in terms of serving as a general pretraining dataset. We leave this study to future work.
>
> We’d also like to emphasize that our goal is not to beat the state-of-the-art in this single piece of work. Existing specialized systems have many years of research (tricks and optimization) built in, so it is not always possible to beat SOTA with an entirely new approach in a single try. The main message from our comparison with baselines is that, even compared to these very strong baselines (SOTA as of a few years ago), the proposed approach, without the specialization used in the baseline techniques, can still achieve a similar level of performance, showing this can be a promising future direction.
>
> *How does multitask performance vary if we pretrain the model on other tasks than object detection?*  We only pretrain on the object detection task (with Object365), so we are unfortunately not able to provide additional evidence on how other pretraining tasks affect these downstream tasks. We conjecture that large-scale image-text pretraining would play an important role, and our methodology would show its real potential with scaled-up pretraining. We believe this is an important question, but also a resource demanding question that requires  further investigation. This work serves as an important stepping stone in that direction.
>
> **Novelty and motivation behind unified interface/model**
>
> Our work builds upon pix2seq, so the framework is the same, and fundamental ideas (e.g., architecture and loss) are similar.  But this  does not mean this work is incremental. Notably, the original pix2seq work only demonstrates that sequence modeling works for object detection, a single vision task. In this work, we propose concrete extensions, and demonstrate this can work for a diverse set of representative vision tasks, covering low, mid, and high-level visual understanding.
>
> Extending the original pix2seq model to image captioning is straight-forward, but it is not trivial to apply it to instance segmentation (which requires dense semantic masks), or to keypoint detection. More importantly, we further solve challenges in joint training of multiple tasks and demonstrate that the same sequence modeling framework can lead to competitive results in multiple “representative/core” vision tasks. As an example, one may say GPT-n is just incremental on top of GPT-1 for the reasons that they all use the same autoregressive transformer model to train language models of different sizes, but clearly they are not incremental. We believe this work is an important step toward a unified model for computer vision.
>
> The reviewer also asks about the benefits of a unified interface/model. Specialized systems can work very well in a pre-defined domain, but they can be difficult to generalize to new tasks (without specific modification of model and re-training). A unified approach may look worse in performance at the beginning (compared to specialized systems), but it takes time to improve them, and we believe it is a necessary step towards a general intelligence system that can interact with and help people in myriad  ways. An important milestone toward a truly usable generic system is to have the architecture and loss function be  general, which was our goal in this work.  If a (new) task cannot be expressed in the system’s interface and trained with one’s architecture/loss, the system has no chance of generalizing it. In other words, without a unified interface, there is zero chance a model is able to generalize to a diverse set of new tasks - they simply cannot represent them. People may not agree if the specific approach proposed in this work is the right way to go, but we believe it is worthy of exploration. And future work can present new approaches for the interfaces or new interfaces, but the early explorations should not be regarded as not useful.

---

> > ### Author Response · Authors · 2022-08-02
> > **Response to Reviewer KcLQ (continued)**
> >
> > **Task ordering**
> >
> > We only use task ordering for estimating an appropriate weighting for different tasks, and after the task weights are obtained, there is no ordering of tasks in the sense that each batch contains all four tasks (with different loss weighting). For task ordering in estimating the task weights, we roughly follow the task difficulty and dependency to order tasks, so that object detection is first, and instance segmentation is second, as it depends on the object detection.  We briefly shuffled the order of keypoint and object detection (on the third and fourth positions), and there doesn’t seem to be a major difference either way. We conjecture that these two tasks have little dependencies between them.
> >
> > **Others**
> >
> > *How does the current setup compare to other works in terms of training/inference efficiency?* The speed of autoregressive generation is dependent on the number of tokens they generate, and for larger numbers of tokens, they would be slower. Please note we can query the model in parallel for independent predictions (e.g., asking keypoints of multiple person objects in parallel).
> >
> > *Does multitask training as a pretraining step help?* This is an interesting idea but  we haven’t explored it yet. When fine-tuned on a single task, it may provide some extra boost.  But  when fine-tuned on all tasks, this seems equivalent to training longer with a different learning rate schedule so one might not see a major boost in performance

---

### Official Review · Reviewer_X67c · 2022-07-11

**Rating:** 8
**Confidence:** 4
**Soundness:** 4 excellent
**Presentation:** 3 good
**Contribution:** 3 good

**Summary:**

This paper proposed a unified sequence interface (Pix2seq++) for computer vision tasks. The authors extend Pix2seq which can do only object detection to more vision tasks such as segmentation, keypoints, and image captioning and train a single model for 4 different tasks. Pix2seq++ unified the decoder outputs by treating these tasks as sequence prediction tasks. The authors use batch mixing instead of data mixing which implied data augmentation for different tasks. The proposed method achieves state-of-the-art performance on 4 benchmarks.

**Questions:**

- What is the correct format for key points?
- For instance segmentation task, how to make the polygon as a sequence -- is there any order to pick the start point, or is there any constraint if the sequence is too long?
- Could the authors describe the difference between different image augmentation methods for different tasks?
- I wonder what is the authors' insights for different methods on instance segmentation (vq-vae vs. seq)? It will be great to discuss this somewhere in the paper.

**Limitations:**

the authors adequately addressed the limitations and potential negative societal impact

**Strengths And Weaknesses:**

[Strengths]
- The proposed method is the first one that unified "core" computer vision approaches into a sequence and shows it can achieve state-of-the-art performance on these tasks. This has a huge impact on the computer vision community, as the diverse nature of the computer vision tasks leads to diverse architecture designs, and the proposed model unifies/simplifies the design of the decoder. I believe this has a broad impact on vision and unified-model communities.
- The proposed method achieves good performance on 4 of the benchmark compared to existing classical methods.
- The authors explore different sampling strategies and mixing ratios for the tasks in the supplementary.

[Weaknesses]
- The keypoint tokenization is not clear to me. From L69, the paper mentions the key points are represented as a sequence of quantized image coordinates. However, from Figure 1, it seems the output is <label, ymin, xmin>, I wonder which one is the correct format for key points.
- For instance segmentation task, how to make the polygon as a sequence -- is there any order to pick the start point, or is there any constraint if the sequence is too long?
- In section 2.3, the authors mention different image augmentation for different tasks. Could the authors describe the difference between different image augmentation methods for different tasks?
- In table 1, there is no citation for the Transformer-based caption method, and it would be better to report image captioning metrics such as cider score for the tasks.
- For instance segmentation, I have a question and want to hear the author's insight. Some concurrent work uses VQ-VAE to serialize the image output such as (UViM and Unified-IO), however, the proposed pix2seq++ directly uses points. I wonder what is the authors' insights for different methods? It will be great to discuss this somewhere in the paper.

---

> ### Author Response · Authors · 2022-08-02
> **Response to Reviewer X67c**
>
> We thank the reviewer for the feedback.  Below, please find our responses to each of the concerns or questions raised in the review:
>
> **The keypoint and instance segmentation tokenization**
>
> We do simply use “<ymin of kp1, xmin of kp2, …>” as a protocol for tokenizing the keypoints, given that there are a small fixed set of keypoints.  In Figure 1, as an illustration, we also include the label for the keypoint (eg nose) for ease of interpretation.  We can remove this from the figure to avoid this confusion.
>
> For polygon tokenization, each time a polygon is sampled, we randomly pick a starting point, and simply truncate the resulting sequence if it is too long (which happens rarely, in cases with very fine-grained annotations).   We will clarify these details in the revision.
>
>
> **Image augmentation for different tasks**
>
> The main image augmentations we use for object detection, instance segmentation are scale jittering (scaling images randomly without changing aspect ratio, crop a fixed size region randomly, and then pad to the maximum size). For keypoint detection, we crop the region of the person and resize it for training the model. For image captioning, we do not use image augmentation (we just rescale images without changing aspect ratio so that the longer side fits maximum size, and then pad the other side to the maximum size).
>
> We will add the citation for image captioning using transformers (and include the cider scores).
>
>
> **Comments on concurrent methods (UViM and Unified-IO) for segmentation**
>
> Both polygons and dense masks are valid data formats for instance segmentation. We believe both representations have their own merits.  We chose polygons as they are the primary approach to how the data are annotated (i.e., annotators draw polygons around the object contours), and they are much more compact/sparse. In principle, we could also control the granularity of the annotation by having different spacing between points of the polygon. The main drawback of polygons may be that the model needs sufficient training data, or pre-training, to learn that we want the polygon that corresponds to dense instance masks.
> Regarding the VQ-VAE style model, we are somewhat  concerned that the final single decoder step may not be expressive enough compared to iterative generation (e.g., for scoring instances in the object detection task). We will add a discussion of this to the revision.

---

### Meta-Review · Area_Chair_BDda · 2022-08-26

**Recommendation:** Accept
**Confidence:** Certain

**Metareview:**

Four reviewers provided reviews for this submission. Several reviewers felt that the idea to unify core vision tasks into a sequential output format is interesting and an entirely new approach and can have a large impact on how we train vision models in the future. There were a few concerns discussed between reviewers and authors. One concern was the comparison to past works that pre-trained on ImageNet vs the proposed model that was pre-trained on Objects365. The second concern was differentiating this work with Pix2Seq. In my opinion, the authors were able to answer both questions well. Overall, given the positive reviews, novelty of the work, potential to cause a significant shift in the approach of future modeling and discussion, I recommend acceptance.


**Award:**

No

---

### Decision · Program_Chairs · 2022-09-14

Accept